# On the Opportunities and Risks of Examining the Genetics of Entrepreneurship

**DOI:** 10.3390/genes13122208

**Published:** 2022-11-25

**Authors:** Ben Heller, Yaniv Erlich, Dafna Kariv, Yossi Maaravi

**Affiliations:** 1Baruch Ivcher School of Psychology, Reichman University, Herzliya 4610101, Israel; 2Efi Arazi School of Computer Science, Reichman University, Herzliya 4610101, Israel; 3Adelson School of Entrepreneurship, Reichman University, Herzliya 4610101, Israel

**Keywords:** genomics, behavioral genetics, entrepreneurship

## Abstract

Recent accomplishments in genome sequencing techniques have resulted in vast and complex genomic data sets, which have been used to uncover the genetic correlates of not only strictly medical phenomena but also psychological characteristics such as personality traits. In this commentary, we call for the use of genomic data analysis to unlock the valuable field of the genetics of entrepreneurship. Understanding what makes an entrepreneur and what explains their success is paramount given the importance of entrepreneurship to individual, organizational, and societal growth and success. Most of the studies into the genetics of entrepreneurship have investigated familial entrepreneurial inclinations in the form of parent–offspring comparisons or twin studies. However, these do not offer a complete picture of the etiology of entrepreneurship. The use of big data analytics combined with the rapidly growing field of genetic mapping has the potential to offer a more complete picture of the etiology of entrepreneurship by allowing researchers to pinpoint precisely which genes and pathways underlie entrepreneurial behavior and success. We review the risks and opportunities which accompany this endeavor and make the case that, ultimately, prioritizing more research into the genetics of entrepreneurship has the potential to be of value to both science and society.

## 1. Introduction

Genetic sequencing technologies, which seek to determine the order of the four chemical bases across an organism’s entire genome, have developed rapidly in the past two decades seeing improvements in capabilities and drastic cost reduction [1]. This pattern has allowed millions of people to sequence and map out (i.e., locate specific segments of interest, such as genes, regulatory instructions, mutation sites, etc.) their DNA, and occasionally share them online on designated platforms (e.g., [2]), creating big data sets of personal genetic information. Such datasets allow scholars to design research that may transform our knowledge and understanding of human existence. One such domain is the investigation of the genetic basis of various personality characteristics. For example, entrepreneurship scholars have long debated about the nature vs. nurture question of entrepreneurial behavior and success [3]; applying the genetic approach may help shed light on the complex interactions between these two important and complementary components. Nevertheless, these scientific opportunities are not without risks. In the current commentary, we discuss the opportunities and threats of examining the genetic basis of entrepreneurial behavior and success, as a stepping stone towards further research into this domain.

## 2. Genomics, Big Data and Medicine

Advances in big data analysis tools have significantly affected the life sciences due to recent accomplishments in genetics and genome sequencing [4]. Sequencing and mapping the genetic information of organisms has resulted in immense data sets (approximately 3 billion base pairs per genome; [5]), characterized by a vastly greater number of variables. This renders classical supervised statistical tools obsolete in terms of the amount of time it takes to analyze the data [6]. The optimization of data analysis algorithms in the age of big data sets has allowed researchers to tap into this previously mysterious body of genetic information. One example is pharmacogenomics, a type of precision medicine, which entails using a patient’s genetic profile to determine the most appropriate treatment [7].

Additionally, studies using genome-wide data sets have been conducted to better understand the causes of specific disorders. These data sets have become increasingly more available due to the preponderance of online genealogy platforms, which streamline the process of obtaining personal genetic information. An example can be found in DNA.Land, which is an online platform through which users can receive a detailed report not only of their genetic makeup and ancestry but also discover interesting research-based insights as to their physical and wellness traits [2]. This platform has been used to understand the etiology of breast cancer, by measuring the phenotypical characteristics of individuals in the database who have or have survived cancer (e.g., age of diagnosis, tumor characteristics, etc.) and statistically analyzing this information along with their genomic data [8].

## 3. Behavioral Genomics—Opportunities and Risks

In addition to studying the genetics of strictly medical conditions, genome-wide association studies (GWAS) have also been conducted to shed light on the genetic correlates of personality characteristics. The most popular theory of personality characteristics is the five-factor model (FFM) [9], which states that the structure of human personality can be characterized using five broad factors: openness to experience, which refers to an individual’s curiosity, inquisitiveness, and tendency towards intellectually challenging tasks; conscientiousness, which refers to an individual’s sense of duty, self-discipline, and responsibility; extraversion, which indicates an individual’s tendency for sociability, assertive, and energetic behavior; agreeableness, which refers to an individual’s cooperativeness, politeness, and tendency towards social harmony; and finally, neuroticism, which reflects an individual’s tendency towards negative affect such as anger, anxiety, or depression. Numerous studies have assessed the genetic correlates of the FFM. For example, openness to experience was found to have two genome-wide significant single-nucleotide polymorphisms (SNPs) in an intergenic region 135 kb downstream from the *RASA1* gene (location: 5q14.3), and conscientiousness in the brain-expressed *KATNAL2* gene (location 182q21.1; [10]). Following the findings of these early endeavors and with the advent of more sophisticated statistical techniques and increases in sample sizes, recent studies point to a more polygenic understanding of complex behavioral traits [11], emphasizing how not a single, but multiple SNPs across the human genome correlate with said traits. In addition, genetic findings are not limited to broad psychological factors, as studies have found genetic correlates for narrower personality characteristics. Among the traits investigated were impulsivity [12], persistence [13], and risk-taking [14].

Looking into the genetics of psychological factors is crucial for both basic and applied science. Basic psychology science aims to shed light on the causal mechanisms behind the human mind and behavior, both normal and abnormal, and is the bedrock of applied research and knowledge. Uncovering the genetics of personality can increase our knowledge of how normal and abnormal personalities are formed, to what extent and how they are affected by internal or environmental factors, and in what way the genetic correlates of personality are expressed neurologically [15]. Genetic research on personality is thus a promising approach that can transform the way we study, understand, and explain human personality. Additionally, genetic research of personality can have far-reaching applications, given that personality traits predict behavioral, social, and health outcomes such as substance abuse, job performance, and educational attainment [16]. Importantly, genetic influences on personality somewhat coincide with those that influence psychopathologies [17], and recent studies have even managed to establish the connection between genes, neuroanatomical structures, and risky behaviors such as alcohol consumption and smoking [18,19]. Thus, understanding the underlying neurobiological processes of personality and its genetic etiology is crucial for treating and diagnosing psychiatric disorders and promoting positive health, social, and behavioral outcomes.

Nevertheless, this approach is not without dangers and risks, mainly concerning social policy [20]. For example, research on the genetics of intelligence has been previously used to justify the allocation of resources to individuals possessing genes found to be predictive of higher intelligence [21]. Another argument tried to associate genes to crime and violence, arguing that society may benefit by locating potential criminals (presumably based on genetic data) and educating them prior to criminal behavior [22]. Given our limited understanding of gene–environment interactions [23], it seems that hasty decisions based on incomplete and early research may have detrimental effects on society.

## 4. The State of the Genetics of Entrepreneurship

These risks are especially salient when examining factors that significantly affect positive and negative life outcomes, yet could be molded by the environment (i.e., education and experience). Such factors include intelligence, prosocial behavior, creativity, etc. One such factor, and the focus of this commentary, is entrepreneurship. Entrepreneurship has been defined as “… an activity that involves the discovery, evaluation, and exploitation of opportunities to introduce new goods and services, ways of organizing, markets, process, and raw materials through organizing efforts that previously had not existed” ([24], p. 4). In both research and practice, it is well established that entrepreneurship is key to economic growth, success, and even the mere survival of individuals, organizations, and nations [25]. Consequently, entrepreneurship has become a prominent research domain (e.g., [26]).

Besides studying the environmental and societal factors, much entrepreneurship research has focused on individuals’ entrepreneurial skills, capabilities, knowledge, and traits [27]. Over the past decades, researchers have sought to uncover the reasons behind people’s choices to launch business initiatives and become entrepreneurs. Some have focused on situational and environmental factors examining governmental policy [28], social network structures [26], entrepreneurial education [29], or specific aspects thereof. However, others have sought entrepreneurship roots at the individual level, examining personality traits or constructs [30]. These individual-level constructs include specific traits (e.g., proactive personality, [31]), broad personality factors (e.g., five-factor model, [32]), motives (e.g., the need for achievement), and more general attitudes, beliefs (e.g., generalized self-efficacy), and intentions.

In addition to understanding the etiology of entrepreneurial behavior (e.g., venture creation), additional studies have focused on what makes a successful entrepreneur, as not every individual who decides to launch a business initiative is ultimately successful [33,34]. Although many factors which influence who becomes an entrepreneur are also predictive of success [34], there are other individual- (e.g., entrepreneurial expertise and social competence; [35,36]) and environmental-level factors (e.g., the entrepreneurial ecosystem; [37]) which differentiate between successful and unsuccessful entrepreneurs, and are thus a main concern for entrepreneurship research.

As aforementioned, some of the traits, long-term motives, attitudes, and beliefs associated with entrepreneurial behavior and success have also been found to have genetic correlates, such as the five model personality factors, and narrower traits such as innovativeness and risk-taking [14]. In the past two decades, researchers have taken a new approach in answering “What makes an entrepreneur?” and what makes them successful by examining the heritability of entrepreneurship [38,39,40]. Most of the recent endeavors at gaining insight therein did so by investigating familial entrepreneurial inclinations. For example, Nicolaou et al. compared the resemblance in entrepreneurial tendencies among identical twins (who share all of their genetic composition) and fraternal twins (who share half of their genetic material on average) to examine the extent to which genetic factors affected their choice to become entrepreneurs, and found significant heritability—around 50%—along with little effect of family environment and upbringing [39]. This finding was replicated in numerous other studies across multiple countries [40,41,42,43], echoing the vast body of twin study literature detailing significant heritability across a myriad of distinct psychological phenomena, such as the Big Five personality traits, depression, economic preferences, and political orientation [44,45]. Others examined the relationship between parents’ and children’s entrepreneurial inclinations, comparing biological and adoptive parent–child duos [38], and found that approximately one-third of the intergenerational effect that entrepreneurial parents have on their children is accounted for by pre-birth factors. Importantly, this study fails to differentiate between genetic and epigenetic factors in the womb, as both are pre-birth factors. Thus, a more complete understanding of the genomics of entrepreneurship can aid in distinguishing the different heritability components.

On the other hand, when examining the genomic component, recent studies [46] have found no significant genome-wide associations with entrepreneurial behavior (a surprising finding given previous heritability estimates), yet this loss of heritability and inability to locate significant associations may be due to common SNPs [47] with insufficiently significant effects, along with the need for relatively large sample sizes to find these effects [48,49]. Nevertheless, it seems plausible to assume that entrepreneurial behavior and success are at least slightly to moderately genetically heritable, notwithstanding scholars’ inability to identify specific SNPs due to its apparently highly polygenetic nature (for one yet-to-be replicated candidate gene study, see: [50]). Constructing a polygenetic index, which entails constructing a weighted sum of all SNPs found to be significantly related to a specific phenotype (e.g., entrepreneurial behavior) via a GWAS and using it examine the predictive power of the index in a similar population, may aid in this regard. However, attempts using this method in the entrepreneurial sphere have shown little success, as can be seen in the example of van der Loos et al. [46], who found that a polygenic index for entrepreneurship captured only 0.2% of the variance in entrepreneurship. Yet, this result may be a function of the relatively small sample size used in the study, and is expected to improve as sample sizes increase [51]. Another alternative is to use of proxy-phenotype analysis [52], in which rather than searching for the genetic correlates of a phenotype of interest (i.e., entrepreneurship), researchers instead conduct a genome-wide association analysis for a theoretically or empirically related proxy phenotype (e.g., risk-taking; [14]). The SNPs are then tested in independent samples for a relationship with the phenotype of interest as a means of uncovering its genomic basis. Importantly, this requires a preliminary examination of what the proper proxy-phenotype candidates are using feature selection models [53], which entails synthesizing the vast literature of the psychological and behavioral predictors of entrepreneurship [54] with our knowledge of their genomic basis. Nonetheless, more significant limitations of using polygenic indexes and proxy-phenotype analyses are: (1) their results are only of relevance to individuals from the genetic population used in the study [55]; and (2) they cannot indicate one’s absolute likelihood of expressing the phenotype, rather they indicate the relative likelihood [56]. These limitations need to be kept in mind and render this endeavor all the more difficult. All things considered, we have yet to reliably pinpoint precisely which genes and pathways underlie the genetic etiology of entrepreneurship [57].

## 5. Entrepreneurial Genomics: A Promising Field

The value of prioritizing entrepreneurial genomic research, at a time in which classic behavioral research is the gold standard, is substantial for both answering unsolved research questions and guiding social policy. First, this line of research can fill the long-standing lacuna regarding the question “what makes a successful entrepreneur?” [58]. As aforementioned, most personality studies on entrepreneurship have been correlational, and have thus contributed little to a complete understanding of the etiology of entrepreneurship (e.g., [31]). Second, given the nascent status of proper measuring tools in the psychology of entrepreneurship, uncovering the genetics of entrepreneurial behavior and success could potentially aid in providing a more complete measurement of the psychological constructs that would otherwise be difficult to measure. Importantly, this would only be the case when the construct in question is latent and hard to identify using classic behavioral measures (e.g., questionnaires, CVs, etc.), and when the association between genes and the construct is strong enough. A general example, given by Benjamin et al. [44], is that of using specific genetics variants as a more reliable indicator of the taste for fatty foods than classical measures. An entrepreneurship-related example may be found in the latent construct of entrepreneurial potential, which represents the degree to which an individual possesses entrepreneurial qualities [59]. Rather than using only self-reported measures of entrepreneurial potential—which may fail to completely account for the latent psychological construct they are meant to measure—scholars could strengthen their measurements by using a genetic analysis and using it as an additional indicator of this potential [57]. Thus, although genetic analyses cannot completely replace self-report or behavioral measures due to the relatively weak gene-behavior associations found in the literature [60], they can provide additional diagnostic information which may be hard to extract from traditional measures. Third, given the aforementioned fact that not all entrepreneurs are successful ones [33,34], more research into the genomics of entrepreneurship may aid us in discerning which factors uniquely affect the tendency for entrepreneurial behavior and which ones affect ultimate entrepreneurial success. For example, in much the same that we can find a genetic basis for personality factors which predict educational attainment (e.g., conscientiousness); [61,62] and a different genetic basis for educational attainment itself [63], we can also possibly find a genetic basis for entrepreneurial personality factors along with a different genetic basis for success in entrepreneurship itself. Finally, this innovative approach may improve beyond current statistical non-genetic models, providing us with a better understanding of the effects of environmental factors (e.g., by absorbing residual variance in the regression models) and allowing more robust statistical inference [64]. This can, for example, offer insight into whether entrepreneurship has robust and direct genomic markers, or rather the genomic markers which predict entrepreneurial behavior and success are actually more directly related to specific personality factors which, in turn, predict entrepreneurship.

The value of entrepreneurial genetic research also extends to the applied sciences. Mainly, this field seems to hold an initial promise towards planning individual-level interventions and policies [65]. Entrepreneurship is a risky endeavor, in that most new ventures fail, which can lead to negative fiscal and psychological consequences for the entrepreneur [66]. Understanding the genetics of entrepreneurship may aid in channeling the proper interventions (i.e., entrepreneurship training and education) based on an individual’s genetic profile, thereby improving their chances of success [39]. Nevertheless, the use of genetic information to predict individual-level phenomena may be limited, given some of the studies discussed above, which show that all SNPs taken together explain only about 25% of individual variability in entrepreneurial behavior [46], and that the main fruits to be reaped from this field of study pertain mainly to population-level differences across entrepreneurial constructs [67]. This can possibly assist in creating population-level policy-related interventions, especially when taken in consideration with related cultural factors which have genetic correlates (e.g., consumption of alcohol; [68]) or potential founder effects (see [69], for an example).

Nonetheless, and similarly to research on the genetic foundations of general psychological constructs, research on the genetics of entrepreneurship may also have harmful consequences. Most saliently, knowledge regarding the genetic makeup predictive of entrepreneurial success along with increasingly simpler DNA mapping methods may lead to discrimination against individuals who lack “entrepreneurial DNA”. In the funding domain, investors may require investees to undergo DNA testing to guide their decision to invest based on how suitable the investees’ genetic makeup is. In education, teachers may devote more resources to students with greater genetic potential to succeed as entrepreneurs, even to the point of setting genetic criteria of acceptance to entrepreneurial education programs. We believe that these negative consequences can be mitigated if appropriately framed in social contexts, similarly to how the significant heritability of intelligence [70] has not led to discrimination against those with seemingly unfavorable genetics of intelligence. Nevertheless, given the history of eugenics [71] and the negative social impact of pseudoscientific thoughts regarding genetics [72], we urge entrepreneurial genetics researchers to be wary of the possibility of harmful misuses and misunderstandings of their findings.

Finally, although entrepreneurship is beneficial to the growth of societies and organizations, there may be some negative individual-level consequences to having entrepreneurial personalities, which should raise important questions as to whether it is inherently positive. Entrepreneurs are known to be high on risk-taking propensity [54], which could lead to negative life outcomes such as substance abuse and delinquency [73]. Additionally, engaging in entrepreneurial activity has psychological, financial, and social costs, such as loneliness [74], grief over repeated business failures [75], and a decrease in socioeconomic status following said failures [76]. Furthermore, it is still unclear precisely what advantages and disadvantages those with an entrepreneurial personality have compared to those without it in terms of varied and understudied life outcomes. Thus, research into the genetics of entrepreneurship should not view the entrepreneurial personality and its genetic correlates as inherently positive, but rather a complex feature of the human psyche worthy of a more comprehensive understanding, much like other well-researched factors in both normal and abnormal.

## 6. Discussion

The budding science of big data has opened up new opportunities for constructing and analyzing vast datasets of scientifically important information, such as genetic profiles. These genetic profiles have already been studied in various spheres, such as medicine and psychology, yet entrepreneurial success—a crucial contributor to an ever-changing and thriving economy—has been studied mainly using correlational psychological methods. Consequently, the genetic etiology of entrepreneurial behavior and success remains relatively unknown, due to some of the statistical, methodological, and financial obstacles mentioned above, such as the very large and costly sample sizes needed to compensate for the stringent significance levels required to properly interpret GWAS [69]. Nevertheless, there exist multiple relatively resource-friendly strategies at the scientific community’s disposal. These include, for example, the aforementioned proxy-phenotype analysis [52], and genomic structural equational modeling [77], which have the potential to boost gene identification efforts while keeping the resource needs at a minimum.

Finally, although this article focuses on the need to prioritize the study of the genomics of entrepreneurship, the future of science lies in a multi-omnic approach. This approach emphasizes the use of data from multiple levels of analysis (i.e., genomics, epigenetics, proteomics, microbiomics, etc.) and its integration in order to more comprehensively understand biological phenomena [78]. Thus, achieving our goal of a more complete understanding of the biological etiology of entrepreneurship necessarily requires a more holistic multi-omnic approach. Thus, genomics, the most mature of the omics fields, has a significant part to play in this endeavor, and we therefore urge to prioritize its use in entrepreneurship research.

In conclusion, we call for the examination of the genetics of entrepreneurship. To this end, we have reviewed the potential risks and value of this research approach to both basic and applied sciences. We believe that prioritizing in this line of research has the potential to elucidate the complex relationship between the biological and environmental factors that “make” an entrepreneur, not by leaning to one side over the other, but rather by offering practitioners and educators insights on how to best design their “nurturing” of entrepreneurship in the face of “natural” constraints. Thus, we issue a call for action: invest in the science of entrepreneurial genetics and unlock its potential, yet do so with caution.

## Data Availability

Not applicable.

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
