# Peer review of "On the Opportunities and Risks of Examining the Genetics of Entrepreneurship"

_genes, 2022, doi:10.3390/genes13122208_

Round 1

Reviewer 1 Report

In this opinion piece, the authors assess the opportunities and risks of examining the genetics of entrepreneurship. Specifically, the authors argue that the science of entrepreneurial genomics should be prioritized to a greater extent.

In general, the opinion piece is timely, relevant, and interesting. However, I feel that the authors should tone down the article a little. For instance, the authors write: “… investing in the science of entrepreneurial genomics should be prioritized to a greater extent, as it holds promise for both research and society.” This assessment is clearly an opinion, which might not be shared among the larger scientific community. Clearly, diseases and maladaptive behaviors that cost billions and billions of dollars every year would be prioritized by many of our scientific colleagues. Although, I am sympathetic with the general notion that entrepreneurship is important for the progress of our societies, I would strongly recommend leaving it up to the reader whether this branch of science should be prioritized or not. That is, I would highlight the pros and cons of this line of research, and only in the conclusion would call to prioritizing this line of research. As it is written now, the authors repeat this demand throughout the text multiple times, which comes across a little pushy.

Additionally, I think the authors should be more precise about the distinction between successful entrepreneurship and unsuccessful ones. I think it would be interesting whether there are certain genetic factors that would favor one’s entrepreneurial success.

More General comments:

-       Lines 67-86: The authors name two regions, i.e. RASA1 gene and the KATNAL2 gene. Now, while this finding was reported in the original paper, in recent years it became evident that almost all behavioral traits (e.g. risk, personality, sexual orientation, IQ) are polygenic. That is, not a single SNP, but a bouquet of SNPs across the whole genome correlate with those behaviors. Therefore, I would recommend to emphasize in this paragraph the polygenic nature of those associations. Similarly, with increasing data sample sizes, it is expected to find many many more associations that correlate with the big five. It is fine to mention the single genes in the text, but it should be clear to the reader, and I would mention this first, that many gene-behavior associations are polygenic.

-       Lines 95-96: While I agree that gene-behavior studies are important and will advance our understanding of personality traits (such as risk, or the big five), I disagree that genetic analyses will replace questionnaires. Why? Because gene-behavior associations are usually weak. Even the best polygenic risk scores will show only a weak correlation with behavior. Thus, even if a polygenic risk score can be constructed for a behavioral trait, it won’t be usable as a diagnostic tool to assess a single persons personality, because the correlation is too weak for this (usually r < .1). Thus, as of now, I don’t see how a genetic test might be able to replace any questionnaire to assess an individuals personality.

-       Line 99-100: the authors cite quite old reference on this topic. In recent years, there were enormous advances, both in sample size and in predictability of such behaviors. Additionally, most recent work was able to link neural features (brain anatomy, function) to genes and the resulting behavior. For reference, see “Aydogan, G., Daviet, R., Karlsson Linnér, R., Hare, T. A., Kable, J. W., Kranzler, H. R., ... & Nave, G. (2021). Genetic underpinnings of risky behaviour relate to altered neuroanatomy. Nature Human Behaviour, 5(6), 787-794.” and Daviet, R., Aydogan, G., Jagannathan, K., Spilka, N., Koellinger, P. D., Kranzler, H. R., ... & Wetherill, R. R. (2022). Associations between alcohol consumption and gray and white matter volumes in the UK Biobank. Nature communications, 13(1), 1-11.

-       Line 220 and 240 are the very same argument. I would reduce and cut redundant statements like those.

-       I think the authors should focus more on the question of what makes a successful entrepreneur, instead of just looking at the driving factors behind founding a company. In general, it would be interesting to see to what extend genes would predict success in this realm, and whether certain traits would support this success.

Reviewer 2 Report

Very interesting work! Thanks for your effort to this field. I do agree this is a very complicated task, since there is no simple direct linear correlation between genetic variations and entrepreneurship. There are so many other factors between them, as you mentioned, Five-Factor model, motives, and so on so forth.

In this commentary, you also mentioned proxy-phenotype method showed slightly better results comparing to polygenic index. I guess this might be the correct path, but we do need figure out what are the true proxy-phenotype candidate, which means we may need other models (feature selection models) to measure the correlation between these proxy phenotypes and entrepreneurship.

And another important question is that to what extent the genetics affect the entrepreneurship? What if genetics only affect some of the personality traits, like Five-Factors and motives, and the entrepreneurship is only the outcome of these 'secondary' personality traits? In a simple model:

genes -> personality traits -> entrepreneurship

In that case, we need totally different model to connect genetics and entrepreneurship
